# Time-series transcriptomic analysis reveals novel gene modules that control theanine biosynthesis in tea plant (*Camellia sinensis*)

**Haisheng Cao[1]☯, Xiaolong He[2]☯, Jinke Du[1]☯, Rui Zhang[3], Ying Chen[1], Yong Ma[3], Qi Chen[1], Congbing Fang[1], Chi-Tang Ho[4], Shihua Zhang●[5]\*, Xiaochun Wan[1]\***

**1** State Key Laboratory of Tea Plant Biology and Utilization, Anhui Agricultural University, Hefei, China, **2** School of Sciences, Anhui Agricultural University, Hefei, China, **3** College of Information and Computer Science, Anhui Agricultural University, Hefei, China, **4** Department of Food Science, Rutgers University, New Brunswick, New Jersey, United States of America, **5** College of Life Science and Health, Wuhan University of Science and Technology, Wuhan, China

☯ These authors contributed equally to this work.
\* zhangshihua@ahau.edu.cn (SZ); xcwan@ahau.edu.cn (XW)

**Data Availability Statement:** All relevant data are within the manuscript and its Supporting Information files.

## Abstract

Theanine (thea) is a unique non-protein amino acid in tea plant (*Camellia sinensis*) and one of the most important small molecular compounds for tea quality and health effects. The molecular mechanism that maintains thea biosynthesis is not clear but may be reflected in complicated biological networks as other secondary metabolites in plants. We performed an integrative transcriptomic analysis of tea seedlings bud and leave over the time-course of ethylamine (EA) treatment that activated thea pathway. We identified 54 consistent differentially expressed genes (cDEGs, 25 upregulated and 29 downregulated) during thea activation. Gene Ontology (GO) functional enrichment analysis of upregulated genes and downregulated genes showed that they may function as a cascade of biological events during their cooperative contribution to thea biosynthesis. Among the total cDEGs, a diversity of functional genes (e.g., enzymes, transcription factors, transport and binding proteins) were identified, indicating a hierarchy of gene control network underlying thea biosynthesis. A gene network associated with thea biosynthesis was modeled and three interconnected gene functional modules were identified. Among the gene modules, several topologically important genes (e.g., *CsBCS-1*, *CsRP*, *CsABC2*) were experimentally validated using a combined thea content and gene expression analysis. Collectively, we presented here for the first time a comprehensive landscape of the biosynthetic mechanism of thea controlled by a underling gene network, which might provide a theoretical basis for the identification of key genes that contribute to thea biosynthesis.

## Introduction

Theanine (thea) is a non-protein amino acid that was first discovered in the leaves of tea plant (*Camellia sinensis*) [1]. It has also been found in trace amounts in two other *Camellia* plants

**Funding:** This work was funded by grants from Fund of National Natural Science Foundation of China (31301248) and the Natural Science Foundation of Anhui Province (No.1608085MC66). The funder had no role in study design, data collection and analysis, decision to publish, or preparation of the manuscript.

**Competing interests:** The authors have declared that no competing interests exist.

(*C. japonica* and *C. sasanqua*) and one mushroom species, *Xerocomus badius* [2]. As a unique characteristic component in tea plant, thea is an important indicator of tea quality due to its umami taste [3]. Many studies have confirmed a variety of health effects of thea on humans, including promoting memory, promoting concentration to learn and study [4]. It also has anti-obesity and neuroprotective effects, facilitates relaxation by inhibiting the side effects of caffeine, reduces blood-pressure, prevents vascular diseases, and enhances anti-tumor activity [5–10]. In recent years, thea has become a focus bioactive in the development and utilization of functional components in tea due to its numerous physiological functions [11, 12].

Compared with the extensive exploration of thea related to food science and human nutrition, the research of its biosynthesis remains largely lagging because of the fact that thea pathway is unique to tea plant and has no reference in data-rich model plants, such as *Arabidopsis thaliana* and rice [13]. The earliest investigations of thea biosynthesis were mainly concerned on a key thea synthetase (TS), which unfortunately remains unclear because the enzyme activity has not been confirmed *in vivo* tea plant [14–16]. In 2018, the genomic dissection of tea plant combined with targeted gene functional analysis revealed a TS candidate (named CsTSI) that had an evidence of thea synthetase activity [17]. With the popularity of high-throughput transcriptome sequencing (RNA-seq) technology, members of thea related genes (e.g., *GS*, *GOGAT*, *GDH*) have been identified and characterized in terms of tissue expression specificity, abiotic/biotic stress induced expression pattern, and genotype-phenotype association coupled with targeted metabolomics analysis of thea biosynthetic pathway [18–21].

Despite significant progress has been achieved in the identification of several key genes involved in thea pathway, our knowledge of its underlying biosynthetic mechanisms is limited. It is noted that previous efforts lacked possible connections among the above individual genes that may cooperatively contribute to thea biosynthesis together with other unknown genes. Moreover, transcriptional regulation of thea pathway under transcription factors (TFs) has not been explored to date. To our knowledge, gene control network underlies the biosynthesis of a wide variety of plant-specialized (secondary) metabolites, such as anthocyanin [22], lignin [23] and glucosinolate [24], suggesting a highly complex regulatory mechanisms related to plant secondary metabolism.

With the above considerations, we designed a time-series transcriptome sequencing of tea seedlings bud and leave, and identified 54 consistent differentially expressed genes (cDEGs) over the time-course of ethylamine (EA) treatment, which has been demonstrated to promote thea accumulation as its biosynthetic precursor in vivo tea plant [17]. Gene Ontology (GO) functional enrichment analysis for upregulated genes and downregulated genes showed that they may behave cascading and cooperative biological functions dominating thea biosynthesis. Among the total 54 cDEGs, different types of functional genes, including enzymes, TFs, transport and binding proteins, were distributed, suggesting a complicated gene control network involved in thea pathway. Next, We modeled a gene network related to thea biosynthesis via the integration of gene co-expression and protein-protein interactions, and three functionally important gene modules were identified from the network. Importantly, we presented experimental evidences for several key genes identified from the network analysis. Our results highlighted the underlying genes and their interaction patterns involved in thea pathway, providing a valuable platform for the further exploration of molecular mechanisms that drive thea biosynthesis.

## Materials and methods

### Plant materials and treatments

Cutting seedlings of tea plant cultivar 'Shuchazao' (one-and-a-half-year-age) were collected from Dechang Nursery Stock Company in Anhui Province (China). Tea cutting seedlings

were then pre-cultivated for 7 days using the tap water after being exposed in the sun in order to remove the hypochlorite in the water because tea is a class of anti-chlorine plant. Next, we transferred the seedlings into culture vessel filled with hydroponic solution in which Shigeki Konish nutritional ingredient accounted for the half of the whole content [25]. Within about 30 days, the tea seedlings grew up strongly and the fresh root produced. We selected those tea cutting seedlings with the same size and similar growing trend and divided them into groups I and II. The tea cutting seedlings in group I were used as control and cultivated in Shigeki Konishi nutrient solution, while those in the group II were cultivated using the same solution with the addition of 25mM EA. The nutrient solution in the culture vessel was changed once every 3 days, and air pumps were used for pouring fresh air into the nutrient solution for 12 hours every day. Four parallel experiments were performed for each treatment, bud and leaf at the developmental stage 'one bud and one leaf' were mixedly-collected for each treatment at the beginning of cultivation, and then sampled with three biological replicates at day(s) of 0, 1, 3, 6, 9, 12, 18, 24 in April of the year (S2 Fig). The samples collected were frozen immediately in liquid nitrogen and stored at -80°C in a freezer.

## Total RNA extraction, library construction and transcriptome sequencing

Total RNA was extracted from the samples using the RNA Prep Pure Plant Kit (TianGen, Shanghai, China). The integrity and quality of RNA was measured using gel electrophoresis (Thermo Scientific, Waltham, MA, USA) and a NanoDrop-2000 spectrophotometer (Nano-Drop, Wilmington, DE, USA). The cDNA libraries were constructed by the staff at Shanghai DayGene Biotechnology Company (Shanghai, China) and paired-end sequenced using an IlluminaHiSeq™ 2000 platform according to the manufacturer's instructions. The raw sequencing reads were submitted to the National Center for Biotechnology Information Sequence Read Archive under Accession No. SRP271510. Clean reads were obtained from the original sequenced reads using in-house Python scripts by removing adaptor sequences and low quality reads according to the method described in [26].

## Identification of DEGs in thea activation

The Fasta sequence file for *Camellia sinensis* var. *sinensis* (CSS) were retrieved from the International Tea Plant Genome Sequencing Consortium [27]. Based on the reference genome, we used the command *hisat2-build* implemented in hisat2 [28] (version 2.1.0) to build a genome index. All the clean reads for each of the above samples were then mapped to the indexed reference genome using the command *hisat2*. The generated SAM format alignments together with the reference genome GTF annotation files were then fed to HTSeq-Count [29] (version 0.9.1) and in-house Python scripts to quantify the expression level of each tea gene model using the widely-used Reads Per Kilobase Per Million Mapped Reads (RPKM) value. For paired Shigeki Konish nutrition solution control and EA aqueous solution activation, DEGs were defined as two folds of change in abundance of expressed transcript by using edgeR [30] (corrected *p*-value $< 0.05$).

## Screening for cDEGs in thea activation using an integrative statistics model

Consistent DEGs (cDEGs) over the time-course of EA treatment usually represent the underlying molecular drivers in the whole thea activation process. Here, we used the robust rank aggregation (RRA) algorithm [31] with default parameters to identify cDEGs in thea activation. This RRA approach was specifically designed for comparison of ranked gene lists and recognition of overlapping genes in different experimental conditions using an integrative order statistic approach. The R package named RobustRankAggreg for the RRA algorithm was

downloaded from the Comprehensive R Network (http://www.omicshare.com) and used to obtain a list of consistent upregulated genes and downregulated genes in different time point of thea activation for the downstream analysis.

## GO functional enrichment analysis of cDEGs

To dissect the functional relationships among the individual cDEGs, GO functional enrichment analysis was performed. We firstly used the Blast2GO program [32] (version 2.3.5) to conduct GO term functional annotation for each of the above obtained cDEGs with default parameters. The cDEGs in thea activation were separated as upregulated and downregulated groups to conduct gene set functional enrichment analysis using the webserver OmicShare tool (http://www.omicshare.com) with default parameters. The enriched GO terms (hypergeometric $p$-value $< 0.05$) in the three functional categories, including biological process, molecular function and cellular component were visualized using the R package ggplot2 [33].

## Gene network modeling for thea biosynthesis

The modeling of a gene network associated with sustaining thea activation under EA treatment may be a good way to identify the underlying molecular mechanisms that maintain thea biosynthesis from the view of the interacting relationships among all the identified cDEGs. To this end, we obtained the co-expression relationships among the cDEGs from our developed TeaCoN (http://teacon.wchoda.com) [34], that documents genome-wide gene co-expression interactions using a computational inference of a large sample of RNA-seq datasets for tea plant available in the Sequence Read Archive (SRA) of National Center for Biotechnology Information [35]. We then used our previously described interolog method [36] to establish the protein-protein interactions among the cDEGs to supplement the above deduced gene co-expression relationships. Thus, the resulting integrated gene-gene interactions represents a more comprehensive gene interacting network among the cDEGs and then can be used to identify the possible key genes involved in thea biosynthesis.

## Determination of thea content by HPLC and qRT-PCR validation of identified thea-related key genes

To provide the possible evidence of the key genes identified from the network analysis for their involvement in thea biosynthesis, the determination of thea content and quantitative real-time PCR (qRT-PCR) validation of genes were performed. The corresponding time-series tea samples in control and EA activation groups were collected with three biological replicates. Thea was extracted from the samples using the protocol described by our previous work [37], and thea content was measured using a Waters 2695 HPLC system (Waters, USA) equipped with a 2489 ultraviolet-visible detector. The liquid column (Phenomenex Kinetex XB-C18, 1.7 micron, 2.1 mm × 100 mm) was used at a flow rate of 0.2 ml/min. The column temperature was set to 40˚C, and the detection wave length was 195nm. The expression patterns of the identified thenine-related key genes were monitored using a QuanStudio 6 Flex Real-Time PCR Detection System (Applied Biosystems, USA). RNA samples were isolated from samples using the same way described in the above transcriptomic experiments, and gene-specific primers of genes were designed according to the manufacturer's instructions (S1 Table). The housekeeping gene glyceraldehyde-3-phosphate dehydrogenase (*GAPDH*) was used as an internal reference gene, and the relative expression of genes was calculated using the $\Delta C_T$ method [38]. To evaluate the statistical significance of a gene's expresion in control and EA treatment at each time point, one-way ANOVA and a Fisher's LSD test were conducted.

## Results

### Identification of cDEGs in thea activation over the time-course of EA treatment

Compared to the same time point of Shigeki Konish nutrition feeding control, a total of 611, 707, 176, 72, 476, 210, 2177 differentially expressed genes (DEGs) were screened using the R package edgeR [30] (corrected $p$-value $< 0.05$, fold change $> 2$) after 1, 3, 6, 9, 12, 18, 24 day(s) of EA treatment, respectively. Among these DEGs, 324/287, 286/421, 74/102, 37/35, 206/270, 47/163 and 956/1221 were upregulated/downregulated, in the corresponding EA activating time points (S1 Fig). It is obvious that DEGs strikingly decreased and increased in number after 9 and 24 days of EA treatment, respectively, indicating the possible transcriptional reprogramming events during the whole thea activation process. We then used the RRA algorithm (see details in Methods) to identify consistent DEGs (cDEGs) over the time-course of EA treatment. Consequently, a total of 54 cDEGs, with 25 upregulated genes and 29 downregulated genes, were identified (S2 Table). Among all the cDEGs, 13 genes (24.1%) were denoted as enzyme genes using a combinational annotation of KEGG, NR and Swiss-Prot databases. These enzyme genes, such as serine carboxypeptidase, glutamate synthase and glutathione *S*-transferase, were reported to be related to plant secondary metabolism in the previous studies [39–41]. We also found a number of TF genes (e.g., *NAC*, *bZIP*) that may act as regulatory switches in thea biosynthesis. In addition to the structural enzyme genes and regulatory TFs, several transport and binding protein, such as ABC transporter family protein, UDP-glycosyltransferase and polyadenylate-binding protein, are also shown to be involved in thea biosynthesis, indicating a hierarchy of gene control network underlying thea biosynthesis as other secondary metabolisms in plants [42].

### GO functional enrichment analysis of cDEGs in thea activation

After a view of the functional distribution of the cDEGs in thea activation, a concerned question remained to be immediately answered, that is, how these cDEGs function together to confer to thea biosynthesis. To answer this question, we subjected these cDEGs to GO functional enrichment analysis. With the aim of being elaborately depicted in gene functional contribution in thea activation, upregulated genes and downregulated genes were separated from the whole cDEGs for the GO functional enrichment analysis. As shown in Fig 1 and S2 Table, upregulated cDEGs were mainly overrepresented in the GO terms endonuclease, hydrolase, transferase, phospholipase and carboxypeptidase in the molecular function; glycyl-tRNA aminoacylation, phospholipid catabolic process, IMP metabolic process and tRNA aminoacylation in the biological process. Whereas downregulated cDEGs were mainly overrepresented in the GO terms oxidoreductase, transferase, glutamate synthase, iron ion binding and transmembrane transporter in the molecular function; toxin catabolic process, monoterpenoid biosynthetic process, glutathione metabolic process and ammonia assimilation cycle in the biological process. Our results indicated that the majority of these cDEGs were significantly enriched in different enzyme activities and metabolic processes despite of minor differences between upregulated genes and downregulated genes, e.g., upregulated genes tend to have the biological function as hydrolase and transferase, and downregulated genes instead tend to have the biological function as transporter and symporter, indicating that upregulated genes and downregulated genes may behave as a cascade of biological events during their cooperative contribution to thea biosynthesis.

（A）

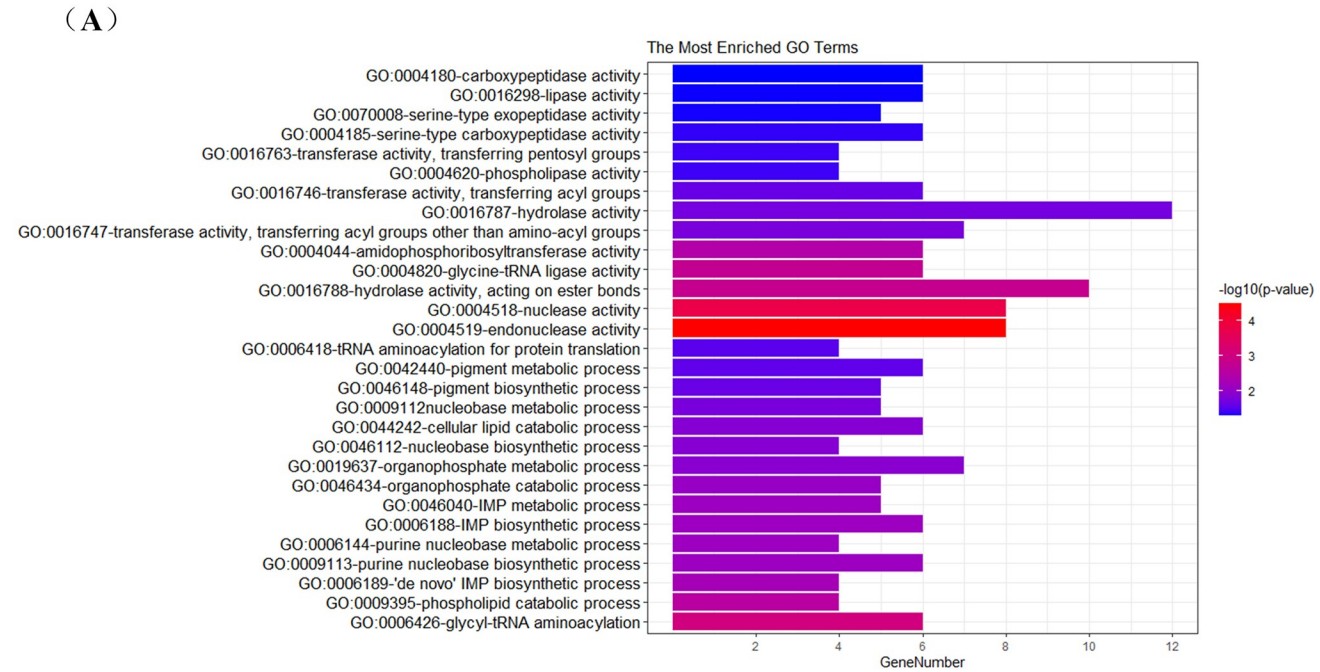

（B）

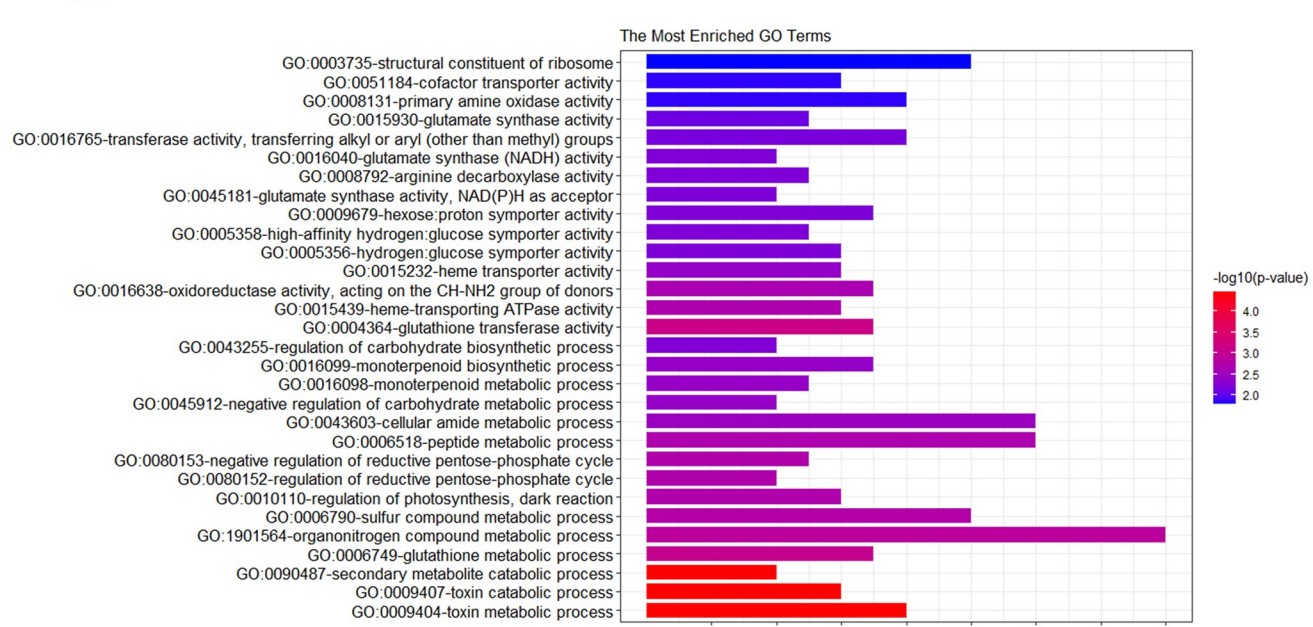

**Fig 1. The most enriched terms in the GO analysis for upregulated genes (A) and downregulated genes (B).** In the diagram, the most enriched GO terms were left-listed, and GeneNumber denoted the number of genes inupregulated/downregulated cDEGs (as a gene set) that have the corresponding GO functional annotation. The right color gradient represented -log10(p-value) of upregulated/downregulated cDEGs enriching in each GO term.

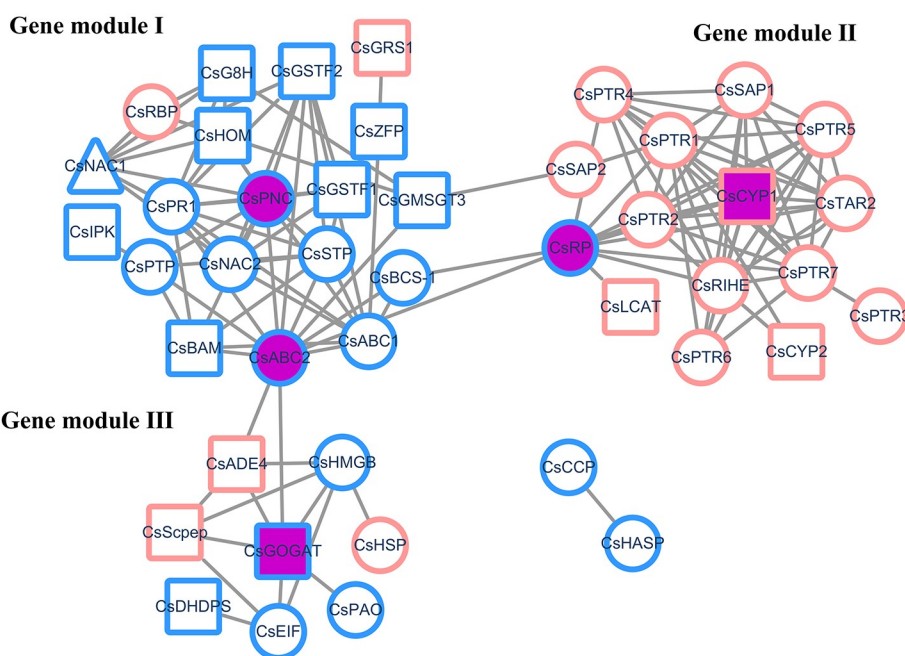

**Fig 2. A modeled gene network associated with thea biosynthesis.** In the network, square, triangle, and circle nodes denoted enzymes genes, TF genes and other types of functional genes, respectively. An edge was placed between two genes indicating they had gene co-expression or protein-protein interactions. The outlined red and blue of a node denoted upregulated and downregulated expression pattern of a certain gene, respectively. Solid purple of a node denoted the five representative hub genes. Module I- III were labeled in the upper-left or upper-right corner of the corresponding gene functional modules.

## Gene network analysis of thea biosynthesis

The modeled gene network was visualized using an open source software platform named Cytoscape [43]. As indicated in Fig 2, the gene network contained 127 non-redundant interactions among 45 (83.3%) genes of the total cDEGs, with three large connected components (overlapped among them) and two isolated genes (*CsCCP* interacts with *CsHASP*). In this network, we found that only one gene pair has both gene co-expression and protein-protein interactions (*CsG8H* interacts with *CsHOM*), which represented the strong gene-gene interaction involved in thea biosynthesis. GO functional enrichment analysis of the three components as different gene sets showed that they have the possible function of transferase activity, hydrolase activity, and organonitrogen compound biosynthesis, respectively (Table 1). From the network perspective, we called these three network components as gene functional modules that may interact with each other as individual gene functional groups to contribute for thea biosynthesis. The overlapping gene nodes among the three gene functional modules were identified as *CsSAP2*, *CsRP*, *CsBCS-1*, *CsABC2*, and *CsADE4* (thereafter named as module-connecting genes). It is noted that only one of the five genes was attributed as enzyme gene, indicating that several other types of functional genes may be the main switch gene nodes in the gene network that maintains thea biosynthesis. In a network, hubs denotes gene nodes with high connectivity (degree) linking to other gene nodes, and degree can be considered as a predictor of essentiality in the network [44]. Table 2 listed the five typical hub genes (solid purple nodes in the network) in the three gene functional modules, as *CsPNC*, *CsABC2*, *CsRP*, *CsCYP1*, and *CsGOGAT*, which represented functionally important genes in the gene network associated with thea biosynthesis. It is noted that *CsRP* and *CsABC2* behaved as both the module-connecting

**Table 1. GO functional enrichment analysis of the three gene functional modules (*p*-value < 0.01).**

| Module ID | # of genes (annotated*) | Possible biological function | GO ID |
|---|---|---|---|
| ModuleI | 19(17) | Transferase activity | GO:0016765 |
| | | Glutathione transferase activity | GO:0004364 |
| | | Peptide metabolic process | GO:0006518 |
| ModuleII | 16(6) | Hydrolase activity | GO:0016787 |
| | | Endonuclease activity | GO:0004519 |
| ModuleIII | 8(7) | Organonitrogen compound metabolic process | GO:1901564 |
| | | Organonitrogen compound biosynthetic process | GO:1901566 |
| | | Glutamate synthase (NADH) activity | GO:0016040 |

*Number of genes that can be annotated in the corresponding gene functional module

**Table 2. Module-connecting genes and typical hub genes in the three gene functional modules.**

| Gene symbol | Functional description | Module-connecting gene or hub gene | Degree |
|---|---|---|---|
| *CsSAP2* | Senescence-associated protein | Module-connecting | 2 |
| *CsADE4* | Amidophosphoribosyltransferase | Module-connecting | 4 |
| *CsBCS-1* | Mitochondrial chaperone BCS1 | Module-connecting | 3 |
| *CsPNC* | Peroxisomal adenine nucleotide carrier | Hub gene | 11 |
| *CsCYP1* | Cytochrome P450 | Hub gene | 11 |
| *CsGOGAT* | Glutamate synthase | Hub gene | 6 |
| *CsABC2* | ABC transporter 2 | Both of them | 12 |
| *CsRP* | Ribosomal protein | Both of them | 11 |

genes and hub genes. Interestingly, previously confirmed thea related enzyme gene *CsGOGAT* behaved as one of the hub genes identified from the network analysis [45].

## Experimental confirmation of identified thea-related key genes

We chose thea as an indicator metabolite for thea pathway. The change of thea content in the time-series tea samples was observed using High Performance Liquid Chromatography (HPLC). Compared with the control group, thea content in EA activation significantly decreased in the 1st day, and consistently increased from the 3rd day to the 12th day, arriving at the highest value (3.43 times, Fig 3). The 12th day can be considered as an EA activation turning point because thea content increased at the highest on this day and then decreased after this day during EA treatment. The expression patterns of eight topologically important genes (hub genes and module-connecting genes) identified from the network analysis were detected using qRT-PCR quantification. Our results indicated the expression of *CsSAP2*, *CsBCS-1*, *CsPNC*, *CsCYP1*, *CsGOGAT*, *CsABC2* and *CsRP* gradually increased, reaching at the highest on the 6th day, and then began to decrease. The content of thea and the expression of the above seven genes all increased to a highest point and then decreased, with a consistent pattern over the time-course of EA activation. It should be noted that the time point is different for the appearance of thea content and gene expression at a highest value, presenting a possible mechanism that the expression (transcription and translation) of tea genes has the lagging phenotypic effect in thea content due to the spatiotemporal separation in gene-thea interaction.

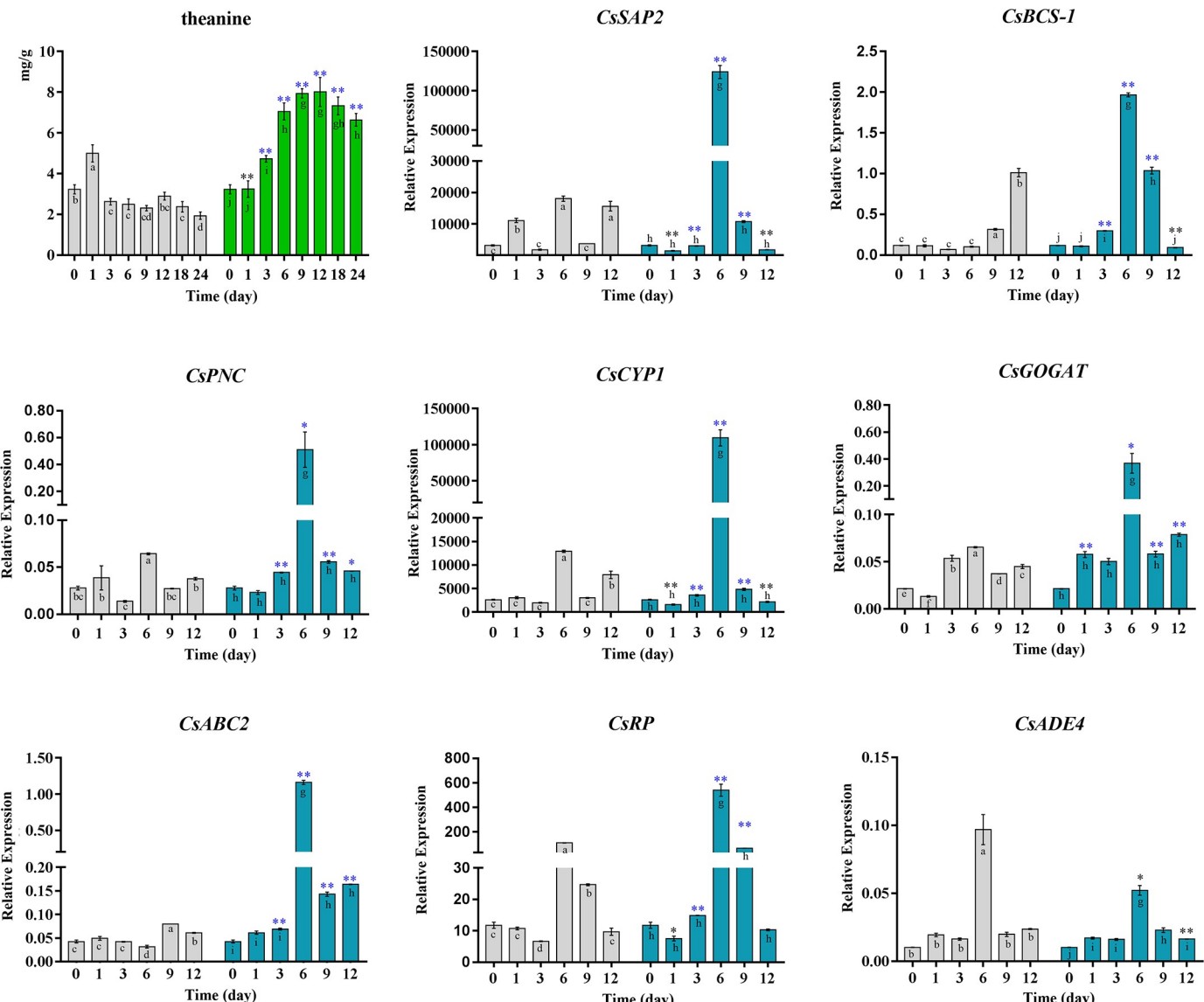

**Fig 3. HPLC and qRT-PCR analysis of thea content and gene expression.** The abscissa indicated samples collected at 0-day, 1-day,3-day, 6-day, 9-day, 12-day, 18-day and 24-day. The grey bar and colored (green, blue) bar represented the changes of thea content or gene expression in the control and EA treatment, respectively. Significant differences comparing the control and EA treatment at each time point using one-way ANOVA and a Fisher's LSD test ($^*p$-value $< 0.05$, $^{**}p$-value $< 0.01$). The black asterisk and blue asterisk denoted a significant drop and increase, respectively. Different letters indicated statistical significance among time points for the control (a, b, c, d, e, f) and EA treatment(g, h, i, j). GraphPad Prism was used to draw the plot, and SPSS was used to conduct statistical analysis.

## Discussion

Plant-specialized metabolites, some of which can be used as flavors, fragrances, colorants, or pharmaceuticals in human life, are biosynthesized through multiple enzymatic steps controlled by the genomes of plants [46, 47]. In the past several decades, functional exploration of enzyme genes in certain plant secondary pathways have been extensively performed, or even dissected informs of biological networks together with other types of functional genes, which enhance our knowledge of the molecular mechanisms of secondary metabolism in plants and give novel clues for applied genetic improvement and metabolic engineering [48]. In this study, we

focused on theanine (thea), a unique small molecular compound in tea plant and an important determinant for tea quality and health effects. We utilized a combinational approach of deep RNA-seq and bioinformatics analysis to uncover the underlying mechanisms of gene network action involved in thea biosynthesis. To this end, we performed a time-series transcriptome sequencing of tea seedlings bud and leave under the feeding of EA aqueous solution, which has been confirmed to be an activator of thea pathway as its biosynthetic precursor [17], and then screened 54 cDEGs over the time course of thea activation. Time-series transcriptome experimental design can help identify a robust set of functional genes related to a certain biological process (e.g., a secondary pathway) by using statistical analysis with the time-course omics data.

Among the total 54 cDEGs, 13 genes were identified as enzyme genes, accounting for a relatively large proportion compared to other types of functional genes. We found that these enzyme genes were attributed to various types of active enzymes, such as ligase, transferase, hydroxylase, synthase and decarboxylase (S3 Table), indicating a complicated and cooperative enzyme-catalyzed process related to thea biosynthesis. We also found that several plant secondary metabolism related enzyme genes, such as serine carboxypeptidase and glutamate synthase. Interesting, glutamate molecular, a precursors of thea, has shown to be catalyzed by the above glutamate synthase and be incorporated into thea pathway [45]. To our knowledge, the biosynthesis of most plant secondary metabolites has been controlled by TF genes through the expression regulation of the related enzyme genes. In this study, we found that *NAC* and *bZIP* were involved in the time course of thea activation. These two TF genes have not been reported to be related to plant-specialized metabolites. We speculated that the transcriptional regulation of thea biosynthesis was specific in tea plant as a unique pathway and this may give useful clues for the possible TF-focused metabolic engineering. It should be noted that a wide variety of functional genes, such as transport and binding protein, were involved in thea biosynthesis, apart from the above structural enzyme genes and regulatory TF genes. All these findings supported our initial hypothesis that a complicated gene control network underlined thea biosynthesis as other plant-specialized metabolites described in the previous studies [22–24].

We then used GO functional enrichment analysis to investigate the functional relationships among the identified cDEGs in thea activation. The total cDEGs were divided into upregulated and downregulated groups according to their expression tendency. The functional comparative analysis of upregulated genes and downregulated genes showed that they may function together in different enzyme activities and metabolic processes conferring thea biosynthesis. Meanwhile, they have the possibility of functioning as a cascade of biological events as separate functional groups. For instance, upregulated cDEGs tend to function as hydrolase and transferase and downregulated cDEGs instead of functioning as transporter and symporter. To explore the interaction pattern among these individual cDEGs, we modeled a gene network related to thea biosynthesis using a combinational index of gene co-expression and protein-protein interaction. From this network, we found three main gene functional modules that may cooperate to contribute for thea biosynthesis. Moreover, several functionally important hub genes and module-connecting genes were computationally identified, and experimentally confirmed using an integrative statistical analysis of HPLC and qRT-PCR qualification for thea content and gene expression, respectively. These findings helped to disclose the underlying functional genes involved in thea pathway and their molecular interaction pattern from the network perspective. Our investigation has significance for the further understanding of molecular regulatory mechanisms that contribute to thea biosynthesis.

## Supporting information

**S1 Table. Primers of eight key genes used for qRT-PCR analysis.**
(XLSX)

**S2 Table. A total of 54 consistent DEGs identified over the time-course of EA activation.**
(XLSX)

**S3 Table. GO analysis of upregulated genes and downregulated genes over the time-course of EA activation.**
(XLSX)

**S1 Fig. Volcano plot that denotes differentially expressed genes (DEGs) in the EA activation.**
(DOCX)

**S2 Fig. Illustration of sample mixing bud and leaf in the one bud and one leaf developmental stage.**
(TIF)

## Acknowledgments

The authors would like to acknowledge the valuable comments and suggestions of the Academic Editor, and the referee. These led to a considerable improvement in the paper.

## Author Contributions

**Conceptualization:** Haisheng Cao, Shihua Zhang, Xiaochun Wan.

**Data curation:** Xiaolong He, Jinke Du, Rui Zhang.

**Formal analysis:** Xiaolong He, Ying Chen, Yong Ma.

**Funding acquisition:** Haisheng Cao, Jinke Du, Yong Ma, Qi Chen.

**Investigation:** Haisheng Cao.

**Methodology:** Xiaolong He, Jinke Du.

**Project administration:** Shihua Zhang.

**Resources:** Ying Chen, Yong Ma.

**Software:** Rui Zhang, Yong Ma.

**Supervision:** Xiaochun Wan.

**Validation:** Haisheng Cao, Jinke Du, Yong Ma.

**Visualization:** Rui Zhang.

**Writing – original draft:** Haisheng Cao, Chi-Tang Ho, Xiaochun Wan.

**Writing – review & editing:** Congbing Fang, Shihua Zhang, Xiaochun Wan.

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
