## [Decision Letter · Decision Letter 0]

30 Jun 2020

PONE-D-20-15564

Time-series transcriptomic analysis reveals novel gene modules that controls theanine biosynthesis in tea plant (Camellia sinensis)

PLOS ONE

Dear Dr. zhang,

Thank you for submitting your manuscript to PLOS ONE. After careful consideration, we feel that it has merit but does not fully meet PLOS ONE’s publication criteria as it currently stands. Therefore, we invite you to submit a revised version of the manuscript that addresses the points raised during the review process.

We look forward to receiving your revised manuscript.

Kind regards,

Tapan Kumar Mondal, Ph.D

Academic Editor

PLOS ONE

Journal Requirements:

Additional Editor Comments (if provided):

I am agree with the reviewer comments. You are encourage to resubmit the manuscript addressing all the poisnt. Also cite some of the classical paper of tea in the related tolpic. Some of them are: Mondal et al., 2004. Plant Cell Tissue Org Cult, 75: 795-856. and Mukhopadhyay et al., (2016) Plant Cell Rep 35(2):255-87

Reviewers' comments:

Reviewer's Responses to Questions

**Comments to the Author**

1. Is the manuscript technically sound, and do the data support the conclusions?

Reviewer #1: Partly

Reviewer #2: Yes

Reviewer #3: Yes

2. Has the statistical analysis been performed appropriately and rigorously? 

Reviewer #1: Yes

Reviewer #2: Yes

Reviewer #3: Yes

3. Have the authors made all data underlying the findings in their manuscript fully available?

Reviewer #1: Yes

Reviewer #2: Yes

Reviewer #3: No

4. Is the manuscript presented in an intelligible fashion and written in standard English?

Reviewer #1: Yes

Reviewer #2: Yes

Reviewer #3: Yes

5. Review Comments to the Author

Reviewer #1: Major comments:

Role of EA should be well mentioned in Introduction.

Line 90- What is meant by sun exposed water?

Line 101-102: One bud and one leaf refers to which leaf (seedling day, young/old, size etc) that should be mentioned. In Thea young leaf should have more theine and this leaf should be consistent in all experiments.

Line 191-192: Number of DEGs decreased and increased after 9 days and 24 days of EA treatment. sentence not clear

Line 248-251: It is not clear why there are different sets of hub and module connecting genes. It appears as if these are created to align the different functional module. A clear and simple explanation is needed to describe Figure 2. There should be one paragraph outlining the functions of these genes in thea pathway and a possible flowchart of heir interaction leading to thea biosynthesis. It is not clear from the network depicted.

Line 261-264: Author has reported that the thea content on EA activation consistently increased from 3rd day to 24th day with 12 th day as turning point. Then he said thea content decreased during the whole EA treatment. Explain your sentence.

In Discussion section, there is no detailing of HPLC and expression studies performed.

Minor comments:

Figure 1 and Figure 3 – Poor resolution and the legends within figure are not clear (eg. theanine content).

Line 91-92 : Detail or Reference of nutritional medium should be provided.

Line 163, 174: Thea and Theanine have been used randomly and in overlapping manner. It is better to stick to one annotation.

Reviewer #2: The present work is well formulated but the design and execution of the experiment is needed to improve and to be more specifically the metabolic pathway enrichment analysis of theanine biosynthesis in tea plants that would be technically sound and novel approach. Some of the remarks I have listed below for improving the paper.

Major remarks:

1) ‘The tea cutting seedlings in group I were used 96 as control and cultivated maintaining the original nutrition solution, while those in the 97 group II were cultivated using the same solution with the addition of 25mM EA 98 aqueous solution to their roots’. How you standardized the concentration of ethylamine (EA) treatment that activated thea pathway?

2) The discussion is too descriptive. The authors did not explain how the differences in EA dependent thea accumulation in tea plant may influence the observed changes in the transcript.

3) For transcriptomics-‘Total RNA was extracted from the samples using the RNA Prep Pure Plant Kit 106 (TianGen, Shanghai, China). The integrity and quality of RNA was measured using 107 gel electrophoresis (Thermo Scientific, Waltham, MA, USA) and a NanoDrop-2000 108 spectrophotometer (NanoDrop, Wilmington, DE, USA)’. For real time PCR- RNA samples were isolated from samples using our previous CTAB method [34].’Why the methods of RNA extraction are different for transcriptomics study and its validation by Real time PCR? Very few genes have been selected for the validation of transcriptomics. Validation of transcriptomics with some more genes are advisable.

4) ‘The housekeeping gene glyceraldehyde-3-phosphate 179 dehydrogenase (GAPDH) was used as an internal reference gene, and the relative 180 expression of genes was calculated using the ΔCT method’. However, it is advisable to use at least 2 or 3 reference genes for achieving accurate normalization.

5) Statistical analysis used here should be more descriptive in method section.

6) Alignment information of sequencing reads and assembly statistics of transcriptome were not discussed and not presented in the table.

7) It is not clear from experiments whether genes like CsBCS-1, CsRP, CsABC2 are regulated by EA treatment or by the accumulation of Thea in plant cell. Interaction between differentially expressed genes should be validated by more than one tool.

8) It would be better to validate the expression of some of the genes by checking the expression of proteins by Western blotting.

9) Fig. 1 in the GO functional enrichment analysis for upregulated (A) and downregulated genes which shown is not clearly visible. Clear Fig. advisable here which represents upregulated (A) and downregulated genes.

10) In Fig. 3 significant difference in expression of a particular gene between control and treated samples are not indicated properly. It is also advised clear fig.

Other remarks:

11) Line no. 73-85 in Introduction section need to change and modify by describing the main aim/objectives of the present study Time-series transcriptomic analysis for theanine biosynthesis in tea plant. Do not present the results in this section.

12) Number of biological replicates of each sample (Control and treated) for transcriptomics are not clear in the method section?

13) The font size of all the figures are not clearly visible.

14) It is important to mention and elaborated description of 3-4 important categories of differentially expressed genes.

15) Line 2 Camellia sinensis should be Italic

16) Line no. 97 EA abbreviation required

17) Line no. 124 Shigeki is repeated

18) Reference no. 26 Journal name, vol. and page no. not mentioned

19) Reference no. 24, 40 and 45 need to correct

20) Scientific name written wherever in reference or main body, make it in italic

21) Line 291-293 is not clear.

22) Line no. 293-296 should be in result section

23) Line no. 321 should be upregulated instead of unregulated

24) The conclusion is not so clear.

25) The grammatical and typing errors should be corrected.

Reviewer #3: There is no mention of the submission of RNA seq data in public databases. Please provide the SRA accession of submission in the manuscript.

Provide better quality figure.

Also comment on other metabolic pathways which get activated with the treatment of EA.

6. PLOS authors have the option to publish the peer review history of their article (what does this mean?). If published, this will include your full peer review and any attached files.

Reviewer #1: No

Reviewer #2: **Yes: **Dr. Bimal Das

Reviewer #3: **Yes: **Atul Kumar Upadhyay

---

## [Author Response · Author response to Decision Letter 0]

16 Jul 2020

Editor Comments to the Author:

I am agree with the reviewer comments. You are encourage to resubmit the manuscript addressing all the poisnt. Also cite some of the classical paper of tea in the related tolpic. Some of them are: Mondal et al., 2004. Plant Cell Tissue Org Cult, 75: 795-856. and Mukhopadhyay et al., (2016) Plant Cell Rep 35(2):255-87

Response to editor

1) I am agree with the reviewer comments. You are encourage to resubmit the manuscript addressing all the poisnt. Also cite some of the classical paper of tea in the related tolpic. Some of them are: Mondal et al., 2004. Plant Cell Tissue Org Cult, 75: 795-856. and Mukhopadhyay et al., (2016) Plant Cell Rep 35(2):255-87

REPLY: Thank the reviewer for the beneficial suggestion. We accordingly cited the related papers. Thanks again.

Review Comments to the Author:

Reviewer #1: Major comments:

Role of EA should be well mentioned in Introduction.

Line 90- What is meant by sun exposed water?

Line 101-102: One bud and one leaf refers to which leaf (seedling day, young/old, size etc) that should be mentioned. In Thea young leaf should have more theine and this leaf should be consistent in all experiments.

Line 191-192: Number of DEGs decreased and increased after 9 days and 24 days of EA treatment. sentence not clear 

Line 248-251: It is not clear why there are different sets of hub and module connecting genes. It appears as if these are created to align the different functional module. A clear and simple explanation is needed to describe Figure 2. There should be one paragraph outlining the functions of these genes in thea pathway and a possible flowchart of heir interaction leading to thea biosynthesis. It is not clear from the network depicted.

Line 261-264: Author has reported that the thea content on EA activation consistently increased from 3rd day to 24th day with 12 th day as turning point. Then he said thea content decreased during the whole EA treatment. Explain your sentence.

In Discussion section, there is no detailing of HPLC and expression studies performed.

Minor comments:

Figure 1 and Figure 3 – Poor resolution and the legends within figure are not clear (eg. theanine content).

Line 91-92 : Detail or Reference of nutritional medium should be provided.

Line 163, 174: Thea and Theanine have been used randomly and in overlapping manner. It is better to stick to one annotation.

Response to Reviewer #1 

1) Role of EA should be well mentioned in Introduction.

REPLY: Thank the reviewer for the beneficial suggestion. Accordingly, we added a description regarding the role of EA on thea pathway as below:

In line of 76-80: 

"we designed a time-series transcriptome sequencing of tea seedlings bud and leave, and identified 54 consistent differentially expressed genes (cDEGs) over the time-course of ethylamine (EA) treatment, which has been demonstrated to promote thea accumulation as its biosynthetic precursor in vivo tea plant [15]."

2) Line 90- What is meant by sun exposed water?

REPLY: We are sorry for the inadequate description regarding "sun exposed water". Here, sun exposed water referred to the tap water after being exposed in the sun in order to remove the hypochlorite in the water because tea is a class of anti-chlorine plant. Accordingly, we re-organized the corresponding sentence as below:

In line of 96-99: 

"Tea cutting seedlings were then pre-cultivated for 7 days using the tap water after being exposed in the sun in order to remove the hypochlorite in the water because tea is a class of anti-chlorine plant."

3) Line 101-102: One bud and one leaf refers to which leaf (seedling day, young/old, size etc) that should be mentioned. In Thea young leaf should have more theine and this leaf should be consistent in all experiments. 

REPLY: We are sorry for the confusing description about the used tea material (mix of bud and leaf, see in Figure S2 in the revised manuscript). One bud and one leaf scientifically represent a stage of tea plant growth, which can be further divided into one bud and one leaf initial development, one bud and one leaf half development, and one bud and one leaf (theanine content keeps highest in this stage than any other tea plant growth stages). In our study, bud and leaf with the same size in the same time points were mixed as the experimental material. Thank the reviewer for pointing this out. Accordingly, we elaborated the corresponding description as below:

In line of 111-114:

"bud and leaf at the developmental stage 'one bud and one leaf' were mixedly-collected for each treatment at the beginning of cultivation, and then sampled with three biological replicates at day(s) of 0, 1, 3, 6, 9, 12, 18, 24 in April of the year (Fig. S2)."

4) Line 191-192: Number of DEGs decreased and increased after 9 days and 24 days of EA treatment. sentence not clear

REPLY: Thank the reviewer for pointing out the confusing description. Now we change as below:

In line of 210-211:

"It is obvious that DEGs strikingly decreased and increased in number after 9 and 24 days of EA treatment, respectively."

5) Line 248-251: It is not clear why there are different sets of hub and module connecting genes. It appears as if these are created to align the different functional module. A clear and simple explanation is needed to describe Figure 2. There should be one paragraph outlining the functions of these genes in thea pathway and a possible flowchart of heir interaction leading to thea biosynthesis. It is not clear from the network depicted.

REPLY: Thank the reviewer for the good suggestion. As the reviewer described, module-connecting genes were created by aligning different functional module. Instead, hub genes were identified according to their network connectivity, which is measured by the edges linking to the genes and thus can be used as a predictor of essentiality in the network. Due to the different strategies, it is logical that there are different sets of hub and module-connecting genes. Regarding the functions of these genes, we think they may cooperatively interact to contribute thea biosynthesis. As we know, thea pathway is a complicated non-linear routes where only a small number of enzyme genes was functionally dissected. Among our screened hub and module-connecting genes, some are enzyme genes, and some are not. Therefore, it is infeasible to illustrate a flowchart of their interactions leading to thea biosynthesis on their non-linear routes, which consist of many chemicals of this pathway. In the revised manuscript (Table), the functional descriptions of these genes were provided. We also elaborated the whole text for Figure 2 and its result description. Thank the reviewer again.

6) Line 261-264: Author has reported that the thea content on EA activation consistently increased from 3rd day to 24th day with 12 th day as turning point. Then he said thea content decreased during the whole EA treatment. Explain your sentence.

REPLY: Thank the reviewer for pointing out the wrong description. Maybe the term 'whole' is misleading, and the 24th day should be changed as 12th day. We intended to show that, during the whole EA treatment process, thea content increased, arriving at the highest on the 12th day, and then decreased. In the revised manuscript, the 24th day was changed as the 12th day. We also delete the term 'whole', and re-organized the sentence as below:

In line of 282-285:

"The 12th day can be considered as an EA activation turning point because thea content increased at the highest on this day and then decreased after this day during EA treatment."

7) In Discussion section, there is no detailing of HPLC and expression studies performed.

REPLY: Thank the reviewer for giving the beneficial suggestion. Accordingly, we added the details in the discussion for the HPLC and expression studies as below:

In line of 354-357:

"Moreover, several functionally important hub genes and module-connecting genes were computationally identified, and experimentally confirmed using an integrative statistical analysis of HPLC and qRT-PCR qualification for thea content and gene expression, respectively."

8) Figure 1 and Figure 3 – Poor resolution and the legends within figure are not clear (eg. theanine content).

REPLY: Thank the reviewer for the good suggestion. We now replaced with high resolution Figure 1 and Figure 3 in the revised manuscript.

9) Line 91-92 : Detail or Reference of nutritional medium should be provided.

REPLY: Thank the reviewer for the good suggestion. Accordingly, we provided a reference regarding the nutritional medium. It sounded more suitable. Thanks again to the reviewer.

10) Line 163, 174: Thea and Theanine have been used randomly and in overlapping manner. It is better to stick to one annotation.

REPLY: Thank the reviewer for the good suggestion. In the manuscript, the chemical theanine was abbreviated as thea for simplification. We think it is a suitable representation. Now it is stick to one annotation. Thank the reviewer again. 

Review Comments to the Author:

Reviewer #2: The present work is well formulated but the design and execution of the experiment is needed to improve and to be more specifically the metabolic pathway enrichment analysis of theanine biosynthesis in tea plants that would be technically sound and novel approach. Some of the remarks I have listed below for improving the paper.

Major remarks:

1) ‘The tea cutting seedlings in group I were used as control and cultivated maintaining the original nutrition solution, while those in the group II were cultivated using the same solution with the addition of 25mM EA aqueous solution to their roots’. How you standardized the concentration of ethylamine (EA) treatment that activated thea pathway?

2) The discussion is too descriptive. The authors did not explain how the differences in EA dependent thea accumulation in tea plant may influence the observed changes in the transcript.

3) For transcriptomics-‘Total RNA was extracted from the samples using the RNA Prep Pure Plant Kit 106 (TianGen, Shanghai, China). The integrity and quality of RNA was measured using 107 gel electrophoresis (Thermo Scientific, Waltham, MA, USA) and a NanoDrop-2000 108 spectrophotometer (NanoDrop, Wilmington, DE, USA)’. For real time PCR- RNA samples were isolated from samples using our previous CTAB method [34].’Why the methods of RNA extraction are different for transcriptomics study and its validation by Real time PCR? Very few genes have been selected for the validation of transcriptomics. Validation of transcriptomics with some more genes are advisable.

4) ‘The housekeeping gene glyceraldehyde-3-phosphate 179 dehydrogenase (GAPDH) was used as an internal reference gene, and the relative 180 expression of genes was calculated using the ΔCT method’. However, it is advisable to use at least 2 or 3 reference genes for achieving accurate normalization.

5) Statistical analysis used here should be more descriptive in method section.

6) Alignment information of sequencing reads and assembly statistics of transcriptome were not discussed and not presented in the table.

7) It is not clear from experiments whether genes like CsBCS-1, CsRP, CsABC2 are regulated by EA treatment or by the accumulation of Thea in plant cell. Interaction between differentially expressed genes should be validated by more than one tool.

8) It would be better to validate the expression of some of the genes by checking the expression of proteins by Western blotting.

9) Fig. 1 in the GO functional enrichment analysis for upregulated (A) and downregulated genes which shown is not clearly visible. Clear Fig. advisable here which represents upregulated (A) and downregulated genes.

10) In Fig. 3 significant difference in expression of a particular gene between control and treated samples are not indicated properly. It is also advised clear fig.

Other remarks:

11) Line no. 73-85 in Introduction section need to change and modify by describing the main aim/objectives of the present study Time-series transcriptomic analysis for theanine biosynthesis in tea plant. Do not present the results in this section.

12) Number of biological replicates of each sample (Control and treated) for transcriptomics are not clear in the method section?

13) The font size of all the figures are not clearly visible.

14) It is important to mention and elaborated description of 3-4 important categories of differentially expressed genes.

15) Line 2 Camellia sinensis should be Italic

16) Line no. 97 EA abbreviation required

17) Line no. 124 Shigeki is repeated

18) Reference no. 26 Journal name, vol. and page no. not mentioned

19) Reference no. 24, 40 and 45 need to correct

20) Scientific name written wherever in reference or main body, make it in italic

21) Line 291-293 is not clear.

22) Line no. 293-296 should be in result section

23) Line no. 321 should be upregulated instead of unregulated

24) The conclusion is not so clear.

25) The grammatical and typing errors should be corrected.

Response to Reviewer #2 

1) ‘The tea cutting seedlings in group I were used as control and cultivated maintaining the original nutrition solution, while those in the group II were cultivated using the same solution with the addition of 25mM EA aqueous solution to their roots’. How you standardized the concentration of ethylamine (EA) treatment that activated thea pathway?

REPLY: Many thanks to the reviewer for this concern. Thea pathway in tea plant was widely studies by isotope labeling (e.g., Sasaoka K et al., Agric Biol Chem,1963, 27: 467-468), and their results showed that the precursors of thea biosynthesis were glutamic acid and ethylamine (EA). According to the results of the previous studies, the optimal concentration of EA was 25 mM. In our study, we set two group for cultivating tea cutting seedlings in hydroponics experiments. The tea cutting seedlings were cultured in Shigeki Konishi nutrient solution as a control group I. Differently, the treatment group II was that the tea cutting seedlings grew in Shigeki Konishi nutrient solution with the addition of 25mM EA. Per the reviewer's concern, we re-organized the description for a clarification as below. Thanks again to the reviewer. 

In line of 103-108:

"We selected those tea cutting seedlings with the same size and similar growing trend and divided them into groups I and II. The tea cutting seedlings in group I were used as control and cultivated in Shigeki Konishi nutrient solution, while those in the group II were cultivated using the same solution with the addition of 25mM EA."

2) The discussion is too descriptive. The authors did not explain how the differences in EA dependent thea accumulation in tea plant may influence the observed changes in the transcript.

REPLY: We are sorry for our inadequate description regarding the aim of our method for EA treatment. As reported in previous studies. EA has been demonstrated to promote thea accumulation as its biosynthetic precursor in vivo tea plant. Therefore, in our study, EA activation of thea pathway was used to identify differentially expressed genes (or transcripts), which should be considered as thea related functional genes. In other words, EA treatment is a tool for studying thea biosynthesis, and EA dependent thea accumulation is a phenotype for observing and explore thea biosynthetic mechanism at molecular level. Maybe, it is not necessary to explain how the differences in EA dependent thea accumulation in tea plant influence the observed changes in the transcript. We hope this is a appropriate response. Many thanks to the reviewer again.

3) For transcriptomics-‘Total RNA was extracted from the samples using the RNA Prep Pure Plant Kit (TianGen, Shanghai, China). The integrity and quality of RNA was measured using gel electrophoresis (Thermo Scientific, Waltham, MA, USA) and a NanoDrop-2000 spectrophotometer (NanoDrop, Wilmington, DE, USA)’. For real time PCR- RNA samples were isolated from samples using our previous CTAB method [34].’Why the methods of RNA extraction are different for transcriptomics study and its validation by Real time PCR? Very few genes have been selected for the validation of transcriptomics. Validation of transcriptomics with some more genes are advisable.

REPLY: We are sorry for our inadequate description regarding the methods of RNA extraction in transcriptomics and real time PCR. In our study, we used the same strategy for the RNA extraction in both high-throughput transcriptomics and low-throughput qRT-PCR experiments. The CTAB method proposed in our previous studies has the same protocols for tea RNA isolation. We now deleted this reference, and re-organized the details description in order to avoid the confusing description. In our study, transcriptomics sequencing, differentially expressed gene screening used the generated transcripomics data, network construction and analysis of differentially expressed genes were integratively utilized to find thea related genes, which could be hub or module-connecting genes from the network topological properties. As in Figure 3, functionally important genes associated with thea biosynthesis have been validated using qRT-PCR expression quantification, relatedness analysis of gene expression and thea accumulation (qRT-PCR, HPLC). Maybe we gave unclear description about this topic. Now we re-organized this point in the whole manuscript for readers. Thank the reviewer again. 

4) ‘The housekeeping gene glyceraldehyde-3-phosphate dehydrogenase (GAPDH) was used as an internal reference gene, and the relative expression of genes was calculated using the ΔCT method’. However, it is advisable to use at least 2 or 3 reference genes for achieving accurate normalization.

REPLY: Thank the reviewer for the good suggestion. We chose GAPDH as the internal reference gene. The first reason is that GAPDH is relatively stable. In addition, we calculated the amplification efficiency of the primers of GAPDH and the target genes, and the amplification efficiency was between 90% and 110%. The melting curve is a single peak, indicating that there is no amplification of non-specific products. All these guaranteed the reliability of our results, and it is also a popular method proposed in current gene expression quantification studies. We will consider the reviewer's suggestion in future attempt by using at least 2 or reference genes to shape a robust experimental system in our lab. Many thanks to the reviewer for giving the valuable idea.

5) Statistical analysis used here should be more descriptive in method section.

REPLY: Thank the reviewer for the good suggestion. During the gene quantification experiments, the significant differences of a certain gene's expression comparing the control and EA treatment at each time point were conducted using one-way ANOVA and a Fisher’s LSD test (*p < 0.05, **p < 0.01). Per the reviewer's concern, we provide more description in the corresponding Method section. Thanks to the reviewer again.

6) Alignment information of sequencing reads and assembly statistics of transcriptome were not discussed and not presented in the table.

REPLY: Thank the reviewer for the concern. In our study, we did not assemble the tea gene modules using the transcriptomic data. The genome of tea plant has been assembled in 2018 by our lab and other collaborative colleagues. In this situation, we conducted a gene expression quantification to screen differentially expressed genes by aligning the sequencing reads to the tea reference genome, by using hisat2, HTSeq-Count, etc. with the default parameters. Per the reviewer's suggestion, we added the related information regarding the analysis in the revised manuscript. Thank much the reviewer again.

7) It is not clear from experiments whether genes like CsBCS-1, CsRP, CsABC2 are regulated by EA treatment or by the accumulation of Thea in plant cell. Interaction between differentially expressed genes should be validated by more than one tool.

REPLY: Thank the reviewer for the concern. The aim of this study is to find the related genes responsible for thea biosynthesis. To this end, EA treatment on tea material was conducted to activate thea pathway because EA is the biosynthetic precursor of thea in tea plant (demonstrated in previous studies). With the EA activation, differentially expressed genes were identified, which can be consider the genes contributing to thea biosynthesis. In other words, these genes were regulated by EA treatment or by the accumulation of thea in plant cell; this is just what the reviewer concerned. Per the reviewer's concern, we elaborated this for readers. In addition, the interaction between differentially expressed genes was shown in gene co-expression and protein network analysis. Thank the reviewer again.

8) It would be better to validate the expression of some of the genes by checking the expression of proteins by Western blotting.

REPLY: Thank the reviewer for the beneficial suggestion. It is important to validate the expression of the possible thea-related genes in protein level by western blotting. As we know, March to May of the year is the best time for tea plant growth and experimental material collection. We regretfully missed it. Usually, it will take 3 to 6 months to prepare the specific antibody for the validation. During the novel coronavirus pneumonia, our laboratory is almost closed with students being kept at home for safety. We are so sorry for that we can't finish the west blotting experiment during the manuscript revision period, and we will attempt if possible in our future study. Many thanks to the reviewer again. 

9) Fig. 1 in the GO functional enrichment analysis for upregulated (A) and downregulated genes which shown is not clearly visible. Clear Fig. advisable here which represents upregulated (A) and downregulated genes.

REPLY: Thank the reviewer for the suggestion. We have now provided a new Figure that is clearly visible. Thanks again to the reviewer.

10) In Fig. 3 significant difference in expression of a particular gene between control and treated samples are not indicated properly. It is also advised clear fig.

REPLY: Thank the reviewer for this concern. Per the reviewer's suggestion, new and more clear Figure was provided. In our study, significant differences in a gene's expression comparing the control and EA treatment at each time point was analyzed using one-way ANOVA and a Fisher’s LSD test (*p < 0.05, **p < 0.01). The black asterisk and blue asterisk denoted a significant drop and increase, respectively. Different letters indicated statistical significance among time points for the control (a, b, c, d, e, f) and EA treatment (g, h, i, j). We elaborated the description in legend of Figure 3 regarding the significant difference in expression of a particular gene between control and treated samples. Thank the reviewer again.

11) Line no. 73-85 in Introduction section need to change and modify by describing the main aim/objectives of the present study Time-series transcriptomic analysis for theanine biosynthesis in tea plant. Do not present the results in this section.

REPLY: Thank the reviewer the good suggestion. We re-organized this section by adding some description regarding the main aim/objectives of the present study. For example, we showed that our results highlight the underlying genes and their interaction patterns involved in thea pathway, providing a valuable platform for the further exploration of molecular mechanisms that drive thea biosynthesis.

12) Number of biological replicates of each sample (Control and treated) for transcriptomics are not clear in the method section?

REPLY: Thank the reviewer for pointing it out. In our study, three biological replicates of each sample for RNA-Sequencing was collected. We now indicated this in the Method section.

13) The font size of all the figures are not clearly visible.

REPLY: Thank the reviewer the good suggestion. Accordingly, we replaced with new figures where the font size is clearly visible.

14) It is important to mention and elaborated description of 3-4 important categories of differentially expressed genes.

REPLY: Thank the reviewer for the good suggestion. The corresponding categories and functional descriptions of these differentially expressed genes were provided in supplemental Tables and in the revised manuscript. Thank very much the reviewer. 

15) Line 2 Camellia sinensis should be Italic

REPLY: Thank the reviewer pointing it out. Now, it is changed as italic form.

16) Line no. 97 EA abbreviation required

REPLY: Thank the reviewer pointing it out. We provided the abbreviation when it first appeared in the manuscript.

17) Line no. 124 Shigeki is repeated

REPLY: Thank the reviewer pointing out the repeated 'Shigeki'. We now deleted it.

18) Reference no. 26 Journal name, vol. and page no. not mentioned

REPLY: Thank the reviewer pointing out the unsuitable reference. We now corrected it.

19) Reference no. 24, 40 and 45 need to correct

REPLY: Thank the reviewer pointing out the unsuitable references. We now corrected them.

20) Scientific name written wherever in reference or main body, make it in italic

REPLY: Thank the reviewer pointing it out. Scientific name is now all in italic. 

21) Line 291-293 is not clear.

REPLY: Thank the reviewer pointing out the unclear description. We now clarified it as below:

In line of 313-316: 

"Time-series transcriptome experimental design can help identify a robust set of functional genes related to a certain biological process (e.g., a secondary pathway) by using statistical analysis with the time-course omics data [45]."

22) Line no. 293-296 should be in result section

REPLY: Thank the reviewer the good suggestion. We now removed it in result section. 

23) Line no. 321 should be upregulated instead of unregulated

REPLY: We correct this misspelling. Thank the reviewer.

24) The conclusion is not so clear.

REPLY: Thank the reviewer the good suggestion. In our study, we conducted the functional enrichment analysis for upregulated and downregulated genes as individual gene groups. Through the comparative analysis of their functions, we found that upregulated genes and downregulated genes may function together in different enzyme activities and metabolic processes conferring thea biosynthesis. Per the reviewer's suggestion, we elaborated the corresponding description as below:

In line of 344-347:

"The functional comparative analysis of upregulated genes and downregulated genes showed that they may function together in different enzyme activities and metabolic processes conferring thea biosynthesis."

25) The grammatical and typing errors should be corrected.

REPLY: Thank the reviewer for the beneficial suggestion. We have now double-edited the whole manuscript to avoid the grammatical and typing errors. Thank the reviewer again.

Review Comments to the Author:

Reviewer #3: 

There is no mention of the submission of RNA seq data in public databases. Please provide the SRA accession of submission in the manuscript.

Provide better quality figure.

Also comment on other metabolic pathways which get activated with the treatment of EA.

1) There is no mention of the submission of RNA seq data in public databases. Please provide the SRA accession of submission in the manuscript.

REPLY: Thank the reviewer the good suggestion. Now we deposited the RNA data of this study in SRA database and also provide the SRA accession of the data in our revised manuscript, as below, in the revised manuscript.

In line of 123-125: 

"The raw sequencing reads were submitted to the National Center for Biotechnology Information Sequence Read Archive under Accession No. SRP271510."

2) Provide better quality figure.

REPLY: Thank the reviewer for giving this concern. Accordingly, all figures are renewed with better quality.

3) Also comment on other metabolic pathways which get activated with the treatment of EA.

REPLY: Thank the reviewer for the good suggestion. As indicated in the manuscript, the activation on thea pathway with the treatment of EA has been demonstrated in previous studies because EA is the biosynthetic precursor of thea in vivo tea plant. No findings regarding the activation on other metabolic pathways (e.g., caffeine and catechins) in tea plant. In this study, theanine (thea) pathway was focused and treated with EA. So we could not giving comments on other metabolic pathways regarding EA treatment. Per the reviewer's concern, we re-organized this background in the revised manuscript for better clear description. Thanks to the reviewer again.

---

## [Decision Letter · Decision Letter 1]

12 Aug 2020

Time-series transcriptomic analysis reveals novel gene modules that controls theanine biosynthesis in tea plant (Camellia sinensis)

PONE-D-20-15564R1

Dear Dr. zhang,

We’re pleased to inform you that your manuscript has been judged scientifically suitable for publication and will be formally accepted for publication once it meets all outstanding technical requirements.

Kind regards,

Tapan Kumar Mondal, Ph.D

Academic Editor

PLOS ONE

Additional Editor Comments (optional):

The paper is written well and hence can be accepted.

Reviewers' comments:

Reviewer's Responses to Questions

**Comments to the Author**

1. If the authors have adequately addressed your comments raised in a previous round of review and you feel that this manuscript is now acceptable for publication, you may indicate that here to bypass the “Comments to the Author” section, enter your conflict of interest statement in the “Confidential to Editor” section, and submit your "Accept" recommendation.

Reviewer #1: All comments have been addressed

Reviewer #2: All comments have been addressed

Reviewer #3: All comments have been addressed

2. Is the manuscript technically sound, and do the data support the conclusions?

Reviewer #1: Yes

Reviewer #2: Yes

Reviewer #3: Yes

3. Has the statistical analysis been performed appropriately and rigorously? 

Reviewer #1: Yes

Reviewer #2: Yes

Reviewer #3: Yes

4. Have the authors made all data underlying the findings in their manuscript fully available?

Reviewer #1: Yes

Reviewer #2: Yes

Reviewer #3: Yes

5. Is the manuscript presented in an intelligible fashion and written in standard English?

Reviewer #1: Yes

Reviewer #2: Yes

Reviewer #3: Yes

6. Review Comments to the Author

Reviewer #1: (No Response)

Reviewer #2: The authors described well all the previous suggested remarks and changed the manuscripts wherever required but it is advisable again to present high resolution Fig. 1, so that it can be clearly readable.

Reviewer #3: The authors have carefully dealt with the issues raised by the reviewers. Thank you. I recommend for the publication.

7. PLOS authors have the option to publish the peer review history of their article (what does this mean?). If published, this will include your full peer review and any attached files.

Reviewer #1: No

Reviewer #2: **Yes: **Dr. Bimal Das

Reviewer #3: **Yes: **Atul Kumar Upadhyay

---

## [Editor Report · Acceptance letter]

17 Aug 2020

PONE-D-20-15564R1 

Time-series transcriptomic analysis reveals novel gene modules that control theanine biosynthesis in tea plant (*Camellia sinensis*) 

Dear Dr. zhang:

I'm pleased to inform you that your manuscript has been deemed suitable for publication in PLOS ONE. Congratulations! Your manuscript is now with our production department. 

Kind regards, 

on behalf of

Dr. Tapan Kumar Mondal 

Academic Editor

PLOS ONE